# GridAR: Glimpse-and-Grow Test-Time Scaling for Autoregressive Image Generation

## Abstract

Recent visual autoregressive (AR) models have shown promising capabilities in text-to-image generation, operating in a manner similar to large language models. While test-time computation scaling has brought remarkable success in enabling reasoning-enhanced outputs for challenging natural language tasks, its adaptation to visual AR models remains unexplored and poses unique challenges. Naively applying test-time scaling strategies such as Best-of-$N$ can be suboptimal: they consume full-length computation on erroneous generation trajectories, while the raster-scan decoding scheme lacks a blueprint of the entire canvas, limiting scaling benefits as only a few prompt-aligned candidates are generated. To address these, we introduce *GridAR*, a test-time scaling framework designed to elicit the best possible results from visual AR models. *GridAR* employs a grid-partitioned progressive generation scheme in which multiple partial candidates for the same position are generated within a canvas, infeasible ones are pruned early, and viable ones are fixed as anchors to guide subsequent decoding. Coupled with this, we present a layout-specified prompt reformulation strategy that inspects partial views to infer a feasible layout for satisfying the prompt. The reformulated prompt then guides subsequent image generation to mitigate the blueprint deficiency. Together, *GridAR* achieves higher-quality results under limited test-time scaling: with $N=4$, it even outperforms Best-of-$N$ ($N=8$) by 17.8% on T2I-CompBench++ while reducing cost by 18.2%. It also generalizes to autoregressive image editing, showing comparable edit quality and a 13.9% gain in semantic preservation on PIE-Bench over larger-$N$ baselines. The code will be publicly released, and our anonymous project page is available at *https://grid-ar.github.io*.

## 1 Introduction

Visual autoregressive (AR) models (Tian et al., 2024; Team, 2024; Sun et al., 2024; Chen et al., 2025) are emerging as a compelling alternative to the long-dominant diffusion paradigm, demonstrating competitive text-to-image generation against landmark models such as DALL·E 3 (Betker et al., 2023) and Stable Diffusion 3 (Esser et al., 2024). By encoding images as sequences of discrete tokens with the aid of VQ-VAE (Van Den Oord et al., 2017), such models operate in a manner akin to large language models (LLMs). Recent work focuses on raster-scan decoding (*i.e.*, line-by-line decoding) and variants, including iterative masking and next-scale prediction (Wang et al., 2024b; Sun et al., 2024; Chen et al., 2025; Liu et al., 2024), while exploring variants such as iterative masked modeling (Xie et al.; Li et al., 2024) and next-scale prediction (Tian et al., 2024; Han et al., 2025). These efforts continue to push the limits of visual fidelity in autoregressive image generation.

As LLM-style raster-scan decoding becomes feasible in text-to-image generation, a natural research question arises: how can test-time computation scaling - shown to enable human-expert level reasoning in language tasks such as math (Wang et al., 2024a) and coding (Chen et al.) - be applied in this setting? These strategies allocate additional computation at inference; for instance, they encourage longer, chain-of-thought (CoT) outputs (Kojima et al., 2022) or employ Best-of-$N$ (Snell et al., 2024) selection with outcome reward model (ORM) to boost the reasoning capabilities of LLMs on cognitively demanding tasks. Despite these successes in language, tailored strategies for visual AR remain underexplored, and it is still unclear how to effectively scale computation or decompose the generation process into multi-steps during test time.

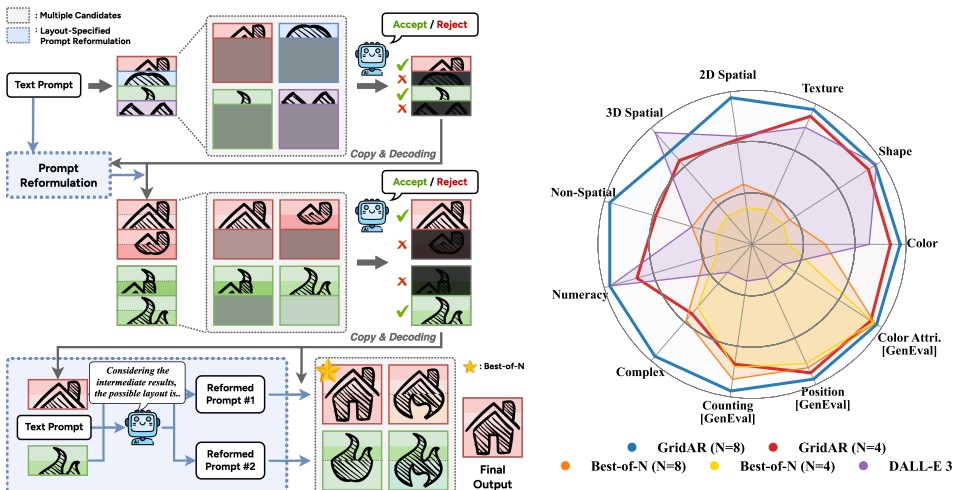

Figure 1: A grid-partitioned progressive generation framework (*GridAR*) for visual AR models. It achieves 17.8% higher text-to-image quality via effective test-time scaling, surpassing Best-of-$N$.

In this paper, we primarily aim to devise a test-time scaling approach for visual AR models, in pursuit of achieving accurate image renderings given complex prompts, including scenarios with multiple objects, spatial relations, and attribute bindings. Prior work mostly ports LLM-style methods (reinforcement learning for token-wise CoT, CoT-augmented prompts) (Jiang et al., 2025) or verifies intermediates in iterative masked AR (Zhang et al., 2025). These approaches suggest that scaling computation at test time can benefit reasoning for image generation in visual AR models; however, they do not fully reflect the unique characteristics that arise when images are generated by AR models.

We highlight two key characteristics of raster-scan image generation for test-time scaling. First, due to the next-token prediction scheme, the model lacks a global blueprint of the full image. For example, when prompted with 'a photo of eight bears,' if the first bear is drawn large in the upper region, the model often leaves the remaining bears undrawn in the lower region. Second, the autoregressive nature makes early errors hard to fix. Consider a prompt requiring four bags: if five handles are already drawn in the upper region, the sequential generation offers no correction. These issues indicate that Best-of-$N$ selection, a representative test-time scaling approach, is not well-suited for visual AR models: once an erroneous trajectory is initiated, it still consumes full computation, and without a global blueprint, wastes resources on misdrawn images.

Building upon this insight, we introduce the *GridAR*, a grid-structured test-time scaling framework for autoregressive image generation. Our approach, inspired by tree-search reasoning in LLMs, focuses computation on regions where further exploration is meaningful and thereby effectively expands the search space. Specifically, the image canvas is partitioned into row-wise tiles and generates multiple candidate images for the same canvas position - *e.g.*, four distinct upper-quarter candidates at the initial stage. Erroneous or infeasible candidates are then rejected, while valid ones are propagated to fill the corresponding canvas positions and serve as anchors that guide the continued generation. This *glimpse-and-grow* strategy guides visual AR models to generate more faithful and sophisticated images that better follow instructions, without requiring any additional training.

One natural artifact of this grid-partitioned progressive generation is a set of partial images, where only the upper portion of the canvas has been rendered. We take these as cues to address the blueprint deficiency in autoregressive models. When a vision-language model serves as a verifier to evaluate grid candidates, we simultaneously perform a *layout-specified prompt reformulation*, in which the prompt is revised to explicitly encode a feasible layout grounded in the observed partial outputs. With this reformulated prompt, we propose two options: (i) apply a three-way classifier-free guidance term to steer the logits to align with the layout specified in reformulated prompts, or (2) directly substitute the prompt in subsequent generation stages, which is cost-efficient. Both approaches guide the model toward a plausible layout consistent with the intermediate results.

Our *GridAR* is thoroughly validated across two tasks: text-to-image generation and image editing, using three models - Janus-Pro (Wu et al., 2025), LlamaGen (Sun et al., 2024), and EditAR (Mu et al., 2025). Extensive experiments show that *GridAR* consistently improves text-to-image generation quality from 4.8% to 17.8% across diverse prompt categories, and even outperforms Best-of-$N$

($N$=8) using only $N$=4 candidates. In image editing, it achieves 17.8% higher semantic preservation compared to a larger-$N$ baseline, demonstrating a more favorable cost-performance trade-off.

In summary, our contribution is threefold:

- We introduce GridAR, a grid-structured progressive generation framework that directs computation toward promising continuations at test time, effectively expanding the search space effectively to elicit the best outputs achievable from visual AR models.
- We propose a layout-specified prompt reformulation that leverages partial views to infer feasible layouts, tackling the blueprint deficiency in autoregressive generation and enriching the candidate pool with prompt-aligned images for more effective test-time scaling.
- Extensive experiments demonstrate that *GridAR* improves generation quality across both text-to-image generation and image editing, drawing out the maximum potential of the pretrained AR model and offering a superior cost-quality trade-off.

## 2 PRELIMINARY

### 2.1 AUTOREGRESSIVE MODELING FOR IMAGE GENERATION

Visual autoregressive (AR) models adapt the next-token prediction paradigm of language modeling to image generation. To enable autoregressive prediction on images, they employ a vector-quantized autoencoder, such as VQ-VAE (Van Den Oord et al., 2017) and VQ-GAN (Esser et al., 2021), which discretizes an image into a finite sequence of codebook indices. Given a trained vector-quantized autoencoder with down-sampling factor $M$ and codebook $\mathcal{Q} = \{e_k\}_{k=1}^{K}$, an image $\boldsymbol{I} \in \mathbb{R}^{H \times W \times 3}$ is encoded into a grid of $h \times w$ latent vectors, where $h = H/M$ and $w = W/M$. These latent vectors are then quantized by the codebook into a discrete sequence $\boldsymbol{x} = (x_1, \ldots, x_N)$, where $x_i \in \{1, 2, \ldots, K\}$ and $N = h \cdot w$.

In text-to-image generation or image editing, the image token sequence $\boldsymbol{x}$ is generated conditioned on context $\boldsymbol{c}$, which may include a text embedding $\boldsymbol{c}_T$ or an image embedding $\boldsymbol{c}_I$. Visual AR models perform next-token prediction modeling on $\boldsymbol{x}$, thereby defining the sequence likelihood as follows:

$$p(\boldsymbol{x} \mid \boldsymbol{c}) = \prod_{n=1}^{N} p(x_n \mid x_{<n}, \boldsymbol{c}).$$

For a visual AR model $p_\phi(\boldsymbol{x} \mid \boldsymbol{c})$ conditioned on context $\boldsymbol{c}$, training on a dataset updates the parameters $\phi$ so that $p_\phi(x_n \mid x_{<n}, \boldsymbol{c})$ is optimized for the distribution of the dataset. During inference, the model generates the sequence by sampling one token at a time, after which a decoder reconstructs the high-resolution image from the corresponding codebook embeddings.

### 2.2 CLASSIFIER-FREE GUIDANCE FOR AUTOREGRESSIVE MODELS

Classifier-free guidance (CFG) (Ho & Salimans, 2022) was first introduced for diffusion models to eliminate the need for an external image classifier while providing a tunable trade-off between fidelity and conditional adherence. At each sampling step, the score estimate is formed as a linear combination of the conditional and unconditional scores. Early intuition suggested that CFG draws samples from a reweighted distribution proportional to $p(\boldsymbol{x} \mid \boldsymbol{c})^{s+1} p(\boldsymbol{x})^{-s}$ (Ho & Salimans, 2022) where $s$ is the guidance scale; however, later analyses show this interpretation is generally incorrect and is better viewed as a predictor-corrector procedure (Bradley & Nakkiran, 2024).

CFG has since been adapted to diverse AR models-such as text-to-image (Sun et al., 2024; Wang et al., 2024b; Wu et al., 2025; Chen et al., 2025; Liu et al., 2024) and text-to-music (Copet et al., 2023)-through logit-space guidance applied before next-token sampling. For an AR model $p_\phi(\boldsymbol{x} \mid \boldsymbol{c})$, to generate $x_i$ at step $i \in \{1, 2, \ldots, N\}$, the guided logits $\hat{l}_i$ are formed as follows:

$$l_i^{\text{sample}} = (1 + s) \cdot l_i^{\text{cond}} - s \cdot l_i^{\text{uncond}} = l_i^{\text{cond}} + s \cdot \left(l_i^{\text{cond}} - l_i^{\text{uncond}}\right),$$

where $l_i^{\text{cond}}$ and $l_i^{\text{uncond}}$ are the conditional and unconditional logits produced by the same model. The next-token distribution can be represented as:

$$p_\phi^{\text{sample}}(x_i \mid x_{<i}, \boldsymbol{c}) = \text{softmax}(l_i^{\text{sample}}) \propto p_\phi(x_i \mid x_{<i}, \boldsymbol{c}) \left(\frac{p_\phi(x_i \mid x_{<i}, \boldsymbol{c})}{p_\phi(x_i \mid x_{<i})}\right)^s.$$

Figure 2: **Visualization of Grid-based Progressive Generation** process in two cases: (a) first-stage rejection (top row), where all candidates are accepted in the second stage; (b) second-stage rejection (bottom row), where all candidates are accepted in the first stage.

Using the autoregressive modeling on $x$, the induced sequence-level sampling distribution satisfies

$$p_\phi^{\text{sample}}\big(\boldsymbol{x} \mid \boldsymbol{c}\big) \propto p_\phi\big(\boldsymbol{x} \mid \boldsymbol{c}\big) \left( \frac{p_\phi\big(\boldsymbol{x} \mid \boldsymbol{c}\big)}{p_\phi\big(\boldsymbol{x}\big)} \right)^s .$$

## 3 TEST-TIME SCALING FOR AUTOREGRESSIVE IMAGE GENERATION

Our main objective is to investigate how test-time computation can be scaled to elicit the best outputs from visual autoregressive (AR) models with next-token prediction. We introduce **GridAR**, a test-time scaling framework that progressively explores generation paths through grid-structured canvases, dynamically directing computation toward promising continuations. Multiple partial image candidates for the same position are generated row-wise for the same canvas position; unlikely ones are pruned early, and viable ones are fixed as anchors to guide subsequent generation (Section 3.1). During this process, we incorporate a layout-aware prompt reformulation alongside candidate verification (Section 3.2). By restructuring the prompt to reflect a feasible layout consistent with selected intermediate results, this step addresses the blueprint deficiency of raster-scan decoding. The reformulated prompt is then used in subsequent decoding - either with three-way classifier-free guidance or simple prompt replacement - to align the remaining regions. These two intertwined components strengthen the candidate pool, leading to more reliable text-to-image generation. An overview of the framework is illustrated in Figure 2.

### 3.1 GRID-BASED PROGRESSIVE GENERATION

In this section, we describe our progressive image completion process through parallel candidate exploration, during which promising candidates are retained - a sophisticated version of Best-of-$N$ for visual AR models. We employ a row-partitioned generation strategy: the first stage uses an $R_1$-row grid to explore $R_1$ initial partial candidates, followed by an $R_2$-row grid for continued generation from the selected anchor candidates. We instantiate $(R_1, R_2) = (4, 2)$; while other configurations are in principle possible, we adopt this setting in this paper.

Starting from this setup, the text-to-image autoregressive model $p_\phi(\boldsymbol{x} \mid \boldsymbol{c}_T)$ begins by generating four distinct candidates ($R_1 = 4$), each corresponding to the upper quarter of the image given the same prompt. Specifically, we represent the canvas $\boldsymbol{x} \in \{1, 2, \ldots, K\}^{h \times w}$ as four contiguous horizontal row segments, $\boldsymbol{x} = \begin{bmatrix} \boldsymbol{x}^{(1)} \\ \vdots \\ \boldsymbol{x}^{(4)} \end{bmatrix}$, where $\boldsymbol{x}^{(r)}$ is the $r$-th row segment and each segment consists of L tokens with $L = \frac{h}{4} \cdot w$. Under this partition, each candidate is autoregressively generated as:

$$p_\phi\big(\boldsymbol{x}^{(r)} \mid \boldsymbol{c}_T\big) = \prod_{n=1}^{L} p_\phi\Big(x_n^{(r)} \Big| x_{<n}^{(r)}, \boldsymbol{c}_T\Big), \qquad r = 1, \ldots, 4,$$

where $x_n^{(r)}$ is the $n$-th discrete latent token index. Different candidates $\boldsymbol{x}^{(r)}$ are generated independently (*i.e.*, without conditioning on each other), while the key-value representations of the prompt $\boldsymbol{c}_T$ are cached once and reused across all rows for computation efficiency. Then the grid-partitioned canvas $\boldsymbol{I}_{\text{grid}}$ containing four candidates is decoded from the vector-quantized embeddings $\boldsymbol{x}^q \in \mathbb{R}^{h \times w \times d}$ (obtained by mapping $\boldsymbol{x}$ to its codebook vectors) through a single forward pass of the decoder $\mathcal{D}_{\text{VQ}} : \mathbb{R}^{h \times w \times d} \to \mathbb{R}^{h \times w \times 3}$ as $\boldsymbol{I}_{\text{grid}} = \mathcal{D}^{\text{VQ}}(\boldsymbol{x}^q)$.

**Candidate verification**  After obtaining image $\boldsymbol{I}_{\text{grid}}$, we assess the four candidate row-segment images *at once* using a verifier $V_\psi$. We here employ a vision-language model as a zero-shot verifier to determine whether candidates are already unlikely to satisfy the given prompt - *e.g.*, when attribute bindings such as color are already incorrect, or when the number of objects exceeds what is required. The verifier $V_\psi$ predicts row-wise judgments directly as:

$$\mathbf{y} = V_\psi(\boldsymbol{I}_{\text{grid}}, \boldsymbol{c}_T) = (y^{(1)}, \ldots, y^{(4)}), \qquad y^{(r)} \in \{\texttt{possible}, \texttt{impossible}\}.$$

It is worth noting that we do not perform top-$k$ selection as in beam search (Sutskever et al., 2014); instead, we reject only those candidates deemed impossible to align with the prompt, while keeping all others. In other words, the number of candidates retained is not fixed [1]. The intuition behind this is that we are evaluating only partial views, where a certain object may simply not appear yet in the visible rows. As a result, top-$k$ selection may prematurely discard many potentially valid candidates for such reasons, thereby harming the sample diversity of the candidate pool. According to the verified results, the set of tokens for four anchor partial images is determined, which will serve as anchors to guide the continued generation. If some candidates are rejected, the rejected one is randomly replaced with feasible one, for example, if $\boldsymbol{x}^{(2)}$ rejected, the anchor set $\boldsymbol{x}_{\text{anchor}}$ can be determined as $\boldsymbol{x}_{\text{anchor}} = [\boldsymbol{x}^{(1)}, \boldsymbol{x}^{(4)}, \boldsymbol{x}^{(3)}, \boldsymbol{x}^{(4)}]^\top$.

**Image expansion from verified candidates**  The discrete token sequences of four verified partial images $\{\boldsymbol{x}_{\text{anchor}}^{(r)}\}_{r=1}^4$ are propagated to the next stage, where two distinct grids with $R_2$-rows are constructed ($R_2 = 2$). In each grid cell, the upper portion is fixed by anchor tokens, and the lower portion is autoregressively continued. Under the $(R_1, R_2) = (4, 2)$ setting, this yields two half-image canvases, each guided by its anchor. Formally, the two half-image canvases are defined as $\tilde{\boldsymbol{x}}_1 = [(\boldsymbol{x}_{\text{anchor}}^{(1)}, \boldsymbol{x}_{\text{gen}}^{(1)}), (\boldsymbol{x}_{\text{anchor}}^{(2)}, \boldsymbol{x}_{\text{gen}}^{(2)})]^\top$ and $\tilde{\boldsymbol{x}}_2 = [(\boldsymbol{x}_{\text{anchor}}^{(3)}, \boldsymbol{x}_{\text{gen}}^{(3)}), (\boldsymbol{x}_{\text{anchor}}^{(4)}, \boldsymbol{x}_{\text{gen}}^{(4)})]^\top$, where each pair consists of a fixed anchor and its autoregressively generated continuation, with $p_\phi(\boldsymbol{x}_{\text{gen}}^{(i)} \mid \boldsymbol{x}_{\text{anchor}}^{(i)}, \boldsymbol{c}_T) = \prod_{n=1}^{L'} p_\phi(x_{\text{gen},n}^{(i)} \mid x_{\text{gen},<n}^{(i)}, \boldsymbol{x}_{\text{anchor}}^{(i)}, \boldsymbol{c}_T)$ with $L'$ denoting the number of tokens in the half-image segment after subtracting anchors.

This process yields four half-image views arranged in a 2-by-2 layout. As in the previous stage, the verifier $V_\psi$ prunes half-image candidates unlikely to satisfy the prompt, with rejected candidates substituted by viable ones. Each verified half-image is then anchored on a new canvas, with the remaining half autoregressively generated to form four complete images. The best image is selected using an output reward model (ORM), analogous to Best-of-$N$ selection. Apart from verification overhead, the number of tokens generated matches that of $N=4$ in standard Best-of-$N$. While our description focuses on $N=4$ for clarity, the framework naturally scales - *e.g.*, using two starting canvases results in $N=8$. Further analysis of rejection rates during progressive generation, along with qualitative examples of rejected samples, is provided in Appendix B.

### 3.2 LAYOUT-SPECIFIED PROMPT REFORMULATION

As described in Section 3.1, our grid-based image generation framework effectively enlarges the search space while circumventing wasted computation on erroneous trajectories. Nevertheless, this pipeline alone is insufficient to ensure a strong candidate pool. Even with carefully selected anchors in the upper half, subsequent decoding may still repeat objects already drawn or omit others required by the prompt. We posit that such failures arise from the absence of a global blueprint: due to the next-token prediction nature of auto-regressive decoding, the model lacks an explicit plan for how the prompt should be realized across the entire canvas. To probe this limitation, we conduct a pilot study that verifies blueprint deficiency as one of the bottlenecks in faithfully portraying

---

[1] In rare cases, all candidates may be rejected; the frequency of such events and our handling strategy are detailed in Appendix B

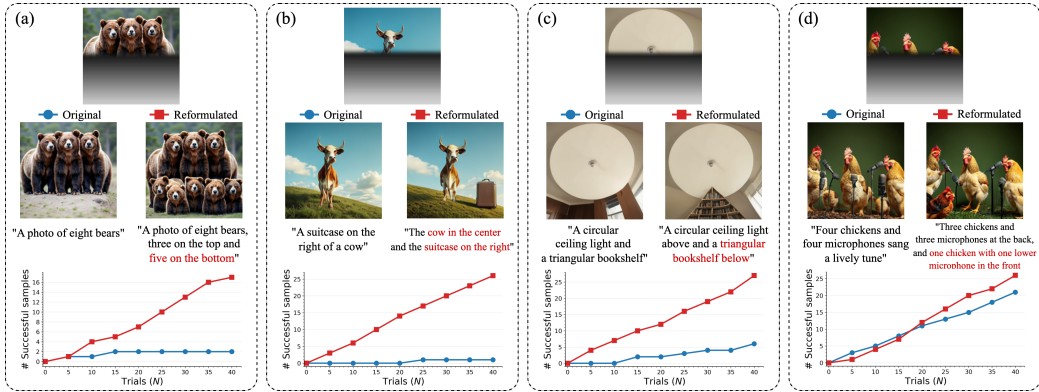

Figure 3: **Motivation of Prompt Reformulation.** Success rate increases significantly with the number of trials when prompt reformulation incorporates a plan for generating lower tokens, rather than relying only on the tokens generated in the upper part.

prompts within visual AR models. This motivates our *layout-specified prompt reformulation*, which dynamically revises the prompt using plausible layouts inferred from the intermediate canvases.

**Pilot study**   We test whether raster-scan decoding, which generates tokens sequentially without global layout, can construct a high-quality candidate pool under test-time scaling, and whether injecting layout knowledge mid-generation can remedy this limitation. Using Janus-Pro 7B (Wu et al., 2025), we focus on prompts where the single-sample setting ($N$=1) fails. As shown in Figure 3, we analyze partially decoded images that remain correct up to the halfway point, conditioning on the upper-half tokens and repeatedly decoding the lower half to measure how many samples eventually satisfy the prompt as trials increase. Results show that many candidates remain incorrect - often duplicating or omitting objects - even under test-time scaling with Best-of-$N$ selection, despite the prompt being achievable (blue curves). Instead of persisting with the original prompt, we reformulate it at the intermediate stage, replacing it with a revised version for subsequent decoding. We revise the prompt by inspecting the intermediate output and specifying a feasible layout that can satisfy the prompt. For instance, in Figure 3 (a), the prompt *"a photo of eight bears"* is reformulated after the partial grid already depicts three bears in the upper region, yielding *"three on top and five on the bottom"*. Interestingly, this simple modification leads to a clear improvement in candidate quality, as shown by the consistently higher success rates as scaling increases (red curves). These results indicate that prompt reformulation allows strong candidate pools to be obtained even with lower levels of scaling.

**Prompt reformulation**   Motivated by this study, we incorporate prompt reformulation into the grid-based progressive generation process. When the verifier $\psi$ evaluates grid candidates, we simultaneously conduct a *layout-specified prompt reformulation*, in which the original prompt is revised to reflect a realizable layout consistent with the observed partial images. The reformulated prompt provides explicit structural cues (*e.g.*, object count or spatial arrangement) inferred from the verified candidates. We consider two alternative strategies: (i) a three-way classifier-free guidance (CFG) that steers logits toward the specified layout by orthogonalizing the reformulated prompt against the original, and (ii) a cost-efficient approach that simply replaces the prompt in subsequent decoding.

**(i) Three-way CFG**   Let $T_u$, $T_o$, and $T_r$ denote the unconditional (null text), original, and reformulated prompts, respectively. At token step $i$, the autoregressive model $f_\theta$ produces a hidden representation, which is projected by the generation head $W$ into the logit space as: $l_i^{(u)} = W f_\theta(x_{<i}, i, T_u)$, $l_i^{(o)} = W f_\theta(x_{<i}, i, T_o)$, $l_i^{(r)} = W f_\theta(x_{<i}, i, T_r)$. We then derive the two directional offsets as $d_{o,i} = l_i^{(o)} - l_i^{(u)}$, $d_{r,i} = l_i^{(r)} - l_i^{(u)}$. To disentangle the layout-specific direction from the original one, we orthogonalize $d_{r,i}$ against $d_{o,i}$, ensuring no interference with the original guidance scale while clearly conveying the layout-specific direction: $\tilde{d}_{r,i} = d_{r,i} - \frac{\langle d_{r,i}, d_{o,i} \rangle}{\|d_{o,i}\|^2} d_{o,i}$.

Finally, the three-way CFG logits are defined as: $l_i^{\text{sample}} = l_i^{(o)} + s_o \cdot d_{o,i} + s_r \cdot \tilde{d}_{r,i}$, where $s_o, s_r \geq 0$ are guidance scales controlling the strengths of the original and reformulated signals. In this work, we do not tune these parameters but simply set $s_o = s_r = s$ to a fixed constant (reusing the conventional scale $s$ for both scales). This formulation preserves the contribution of the original instruction while providing a clear layout-specific signal from the reformulated prompt.

Table 1: T2I-CompBench++ results by dimensions, comparing Best-of-$N$ with GridAR test-time scaling on Janus-Pro and LlamaGen. Scores are reported using metrics proposed in the benchmark.

| Method | Attribute Binding | | | Object Relationship | | | Numeracy | Complex |
|---|---|---|---|---|---|---|---|---|
| | Color | Shape | Texture | 2D Spatial | 3D Spatial | Non-Spatial | | |
| *Diffusion Models* | | | | | | | | |
| SDXL | 0.5879 | 0.4687 | 0.5299 | 0.2133 | 0.3566 | 0.7673 | 0.4988 | 0.3237 |
| Pixart-$\alpha$ | 0.6690 | 0.4927 | 0.6477 | 0.2064 | 0.3901 | 0.7747 | 0.5032 | 0.3433 |
| DALL·E 3 | 0.7785 | 0.6205 | 0.7036 | 0.2865 | 0.3744 | 0.7853 | 0.5926 | 0.3773 |
| SD3 | 0.8132 | 0.5885 | 0.7334 | 0.3200 | 0.4084 | 0.7782 | 0.6174 | 0.3771 |
| FLUX.1 | 0.7407 | 0.5718 | 0.6922 | 0.2863 | 0.3866 | 0.7809 | 0.6185 | 0.3703 |
| *Auto-Regressive Models* | | | | | | | | |
| Lumina-mGPT | 0.6371 | 0.4727 | 0.6034 | - | - | - | - | - |
| Emu3 | 0.6107 | 0.4734 | 0.6178 | - | - | - | - | - |
| LlamaGen | 0.2927 | 0.3160 | 0.3828 | 0.1118 | 0.1510 | 0.7143 | 0.2727 | 0.2445 |
| LlamaGen + BoN (N=8) | 0.5143 | 0.4465 | 0.5850 | 0.1578 | 0.2016 | 0.7543 | 0.4197 | 0.3054 |
| **LlamaGen + GridAR (N=4)** | 0.4969 | 0.4540 | 0.5675 | 0.1466 | 0.1946 | 0.7470 | 0.4118 | 0.2993 |
| **LlamaGen + GridAR (N=8)** | **0.5774** | **0.4783** | **0.5984** | **0.1830** | **0.2019** | **0.7570** | **0.4407** | **0.3103** |
| Janus-Pro | 0.5388 | 0.3476 | 0.4357 | 0.1607 | 0.2806 | 0.7733 | 0.4467 | 0.3796 |
| Janus-Pro + BoN (N=8) | 0.7234 | 0.4178 | 0.5600 | 0.2430 | 0.3165 | 0.7853 | 0.5068 | 0.3926 |
| **Janus-Pro + GridAR (N=4)** | 0.8050 | 0.6014 | 0.7268 | 0.2833 | 0.3503 | 0.7887 | 0.5684 | 0.3905 |
| **Janus-Pro + GridAR (N=8)** | **0.8172** | **0.6174** | **0.7408** | **0.3214** | **0.3587** | **0.7930** | **0.5932** | **0.4041** |

**(ii) Prompt replacement**    As a cost-efficient alternative, we directly substitute $T_r$ for $T_o$ in subsequent decoding steps without modifying the logit computation of classifier-free guidance. Although this strategy does not match the fine-grained signal of a three-way CFG, it offers a lightweight option that still decently guides the model toward layouts consistent with the verified intermediate results. The logit $l_i^{\text{sample}}$ under this strategy follows the standard classifier-free guidance formulation: $l_i^{\text{sample}} = l_i^{(r)} + s_r \cdot d_{r,i} = l_i^{(r)} + s_r \cdot (l_i^{(r)} - l_i^{(u)})$. The same CFG scale $s_r$ used in the earlier image generation with the original prompt is reused here. We show that this strategy outperforms approaches that employ a planner to specify layouts prior to generation in AR models (see Section 4.4).

## 4 EXPERIMENTS

 We conduct a collection of experiments to validate *GridAR* on visual autoregressive (AR) models through two primary tasks: text-to-image generation and image editing. First, we show that our framework can consistently elicit superior image generation results compared to existing test-time scaling methods across diverse prompt categories (Section 4.2). We then demonstrate its versatility in image editing, where the model receives both an edit instruction and a source image, and verify that *GridAR* likewise enhances the effectiveness of computation scaling for this task (Section 4.3). Beyond these benchmark evaluations, we further conduct in-depth analyses addressing research questions raised by our framework, including robustness to different verifiers and comparisons between design choices (Section 4.4). Lastly, we present an ablation study to analyze the contribution of each component and compare two strategies for prompt reformulation (Appendix D.2).

### 4.1 EXPERIMENTAL SETUP

**Implementation details**    We use Janus-Pro-7B (Wu et al., 2025) and LlamaGen (Sun et al., 2024) as backbone models for autoregressive text-to-image generation, and EditAR (Mu et al., 2025) for image editing tasks. Across all experiments, Qwen2.5-VL (Bai et al., 2025) is employed as the outcome reward model for both our method and test-time scaling baselines. For the classifier-free-guidance scale, we set $s_o = 5$ for Janus-Pro, and $s_o = 6.5$ for LlamaGen, following the original paper setup. In *GridAR*, the guidance scale for the reformulated prompt is set equal to the original scale ($s_r = s_o$). To evaluate grid candidates and conduct prompt reformulations, we deploy GPT-4.1 as the verifier $V_\psi$, while other verifier models are also tested in Section 4.4.

**Datasets and evaluation metrics**    Text-to-image generation is evaluated on two benchmarks: T2I-CompBench++ (Huang et al., 2025) and GenEval (Ghosh et al., 2023). T2I-CompBench++ comprises 8,000 compositional prompts across seven categories. Evaluation follows the metrics proposed in the original paper: BLIP-VQA (Color, Shape, Texture), UniDet (2D/3D Spatial, Numeracy), ShareGPT4V (Non-Spatial), and the 3-in-1 score (Complex). GenEval includes over 500 prompts from six categories, and performance is measured by binary correctness on compositional properties, using models such as Mask2Former (Cheng et al., 2022) and CLIP ViT-L/14 (Radford et al., 2021). For image editing, we evaluate on PIE-Bench (Ju et al., 2024), covering 9 editing scenarios and 700 images with paired source images and edit instructions. Performance is assessed along two axes: instruction following and semantic preservation. Instruction following is measured

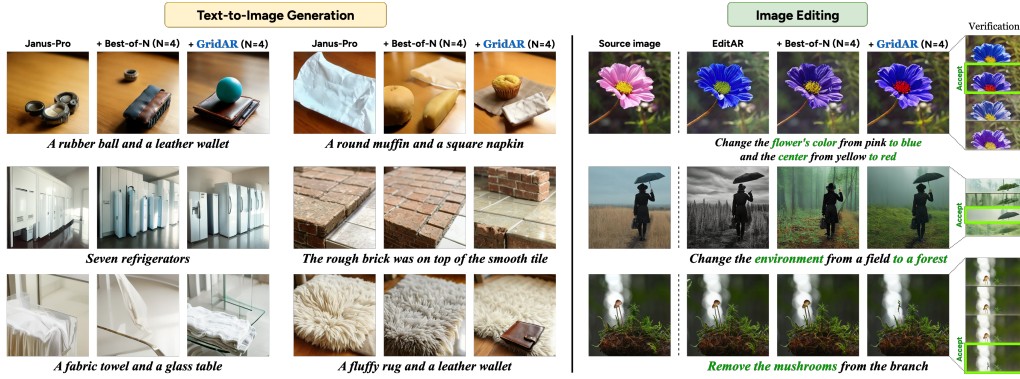

Figure 4: **Qualitative Results** comparing single-generation outputs, Best-of-$N$ ($N = 4$) outputs, and outputs obtained by applying *GridAR* ($N = 4$) on text-to-image generation and image editing.

by the CLIP similarity (Hessel et al., 2021), and source preservation is measured by structure distance with DINO-ViT (Caron et al., 2021) and various perceptual metrics - PSNR, LPIPS (Zhang et al., 2018), MSE, and SSIM (Wang et al., 2004). We primarily compared our method against Best-of-$N$ scaling, and include AR and diffusion-based models as baselines. Evaluation protocols and baseline details are provided in Appendix C.

### 4.2 TEXT-TO-IMAGE GENERATION

We test the effectiveness of *GridAR* in improving image generation quality by scaling test-time compute on text-to-image benchmarks. As shown in Table 1, *GridAR* improves the average score by 17.8% and 4.8% for Janus-Pro and LlamaGen, respectively, under the same $N$ across diverse prompt scenarios. Notably, *GridAR* with $N=4$ even outperforms Best-of-$N=8$ on Janus-Pro, achieving a gain of 14.4% and demonstrating a better cost-performance trade-off (see Section 4.4). These results suggest that our framework is able to derive a higher-quality candidate

Table 2: GenEval results on three selected dimensions and overall.

| Method | Counting | Position | Color Attribution | Overall |
|---|---|---|---|---|
| *Diffusion Models* | | | | |
| SDXL | 0.39 | 0.15 | 0.23 | 0.55 |
| Pixart-$\alpha$ | 0.44 | 0.08 | 0.07 | 0.48 |
| DALL·E 3 | 0.47 | 0.43 | 0.45 | 0.67 |
| SD3 | 0.72 | 0.33 | 0.60 | 0.74 |
| *Auto-Regressive Models* | | | | |
| Show-o | 0.49 | 0.11 | 0.28 | 0.53 |
| Show-o + PARM (N=20) | 0.68 | 0.29 | 0.45 | 0.67 |
| Emu3 | 0.34 | 0.17 | 0.21 | 0.54 |
| Infinity | - | 0.49 | 0.57 | 0.73 |
| LlamaGen | 0.12 | 0.14 | 0.05 | 0.34 |
| LlamaGen + BoN (N=8) | 0.21 | 0.22 | **0.14** | 0.44 |
| **LlamaGen + GridAR (N=8)** | **0.24** | **0.25** | 0.13 | **0.46** |
| Janus-Pro | 0.59 | 0.77 | 0.65 | 0.79 |
| Janus-Pro + BoN (N=8) | 0.76 | 0.86 | 0.72 | 0.86 |
| **Janus-Pro + GridAR (N=8)** | **0.79** | **0.92** | **0.73** | **0.88** |

pool. In particular, when paired with stronger visual AR models such as Janus-Pro, the synergy appears more pronounced-likely due to their improved ability to follow layout specifications and to generate more accurate initial candidates. For the GenEval benchmark, we report performance on Counting, Position, and Color Attribution tasks in Table 2 (other dimensions are already saturated at scores near 0.90-0.99; overall results are provided in the last column). Our method also enhances text-to-image generation quality over Best-of-$N$ across most dimensions. In addition to quantitative results, we present a range of qualitative examples in Figure 4 (left) and Appendix E, illustrating how *GridAR* leads to more accurate and instruction-aligned final selections.

### 4.3 RESULTS ON IMAGE EDITING

We also validate that our test-time scaling framework can be extended naturally to image editing and boost the quality of edited images. To adapt *GridAR* to this setting, we modify the prompt for the verifier to account for both source preservation and adherence to edit instructions

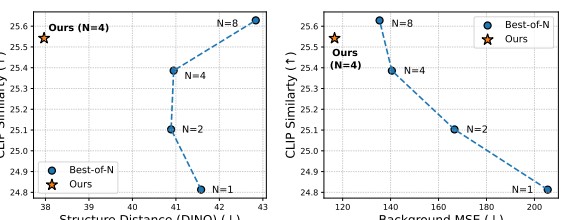

Figure 5: Image editing results on PIE-Bench.

(prompts are provided in Appendix F). Prompt reformulation is applied similarly to text-to-image generation, where feasible layouts are inferred from intermediate results to satisfy the edit instruction. We compare our method against Best-of-$N$ scaling using the same backbone (EditAR), as well as test-time scaled diffusion-based editing models, across 7 metrics. As shown in Figure 5 and Table 4, *GridAR* significantly improves background preservation over Best-of-$N$ ($N=4$), reducing structure-aware distance by 7.27% and MSE by 17.01%. Edit instruction fidelity, measured by CLIP similarity, also shows improvement. Even when compared to Best-of-$N$ ($N=8$), our method achieves comparable CLIP similarity (25.542 vs. 25.628), while substantially outperforming in source preser-

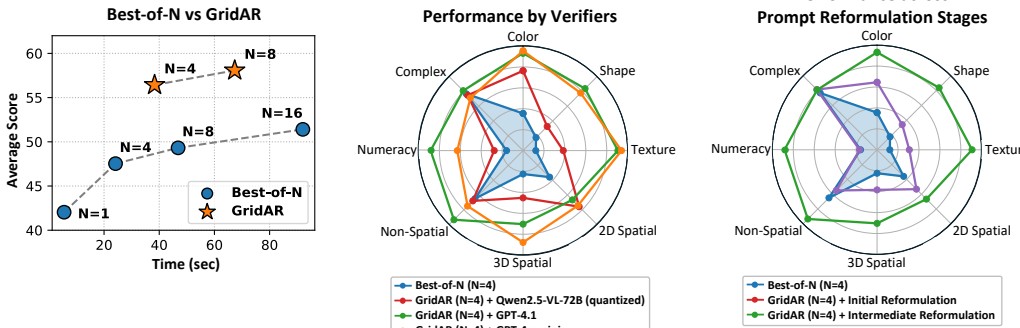

Figure 6: **Left**: Computation-performance trade-off; **Center**: Performance across dimensions by different verifiers; **Right**: Performance across dimensions by different prompt reformulation timings.

vation - showing lower structure-aware distance (37.679 vs. 42.873) and lower MSE (116.637 vs. 135.404). Edited samples are provided in Figure 4 (right), and full quantitative results, including comparisons with test-time scaled diffusion-based models, are reported in Table 4.

### 4.4 IN-DEPTH ANALYSIS

We analyze our framework from three perspectives: (i) the trade-off between computational cost and performance, (ii) comparative performance of different verifier architectures, and (iii) the timing of prompt reformulation. Experiments are conducted under eight dimensions in T2I-CompBench++.

**Cost-performance Pareto analysis** While *GridAR* shows notable improvements over Best-of-*N*, it incurs additional computational cost due to the verification step - though this is substantially reduced by verifying four candidates at once. To better understand the trade-off between performance gains and computational overhead, we conduct the Pareto analysis, as shown in the left plot of Figure 6. Using the Janus-Pro backbone, *GridAR* achieves a 14.4% performance improvement while reducing computational cost by 18.2% compared to Best-of-*N* with *N*=8. Specifically, the single-batch processing time (measured on 2× RTX 3090 GPUs) is 38.31s for *GridAR*, 24.15s for Best-of-*N* (*N*=4), and 46.85s for Best-of-*N* (*N*=8). These results demonstrate that *GridAR* offers a more favorable cost-performance trade-off.

**Comparative evaluation of verifiers** As our framework exploits a verifier to assess partial image views and infer layouts in a zero-shot manner, it assumes a certain degree of image understanding capability from the verifier. In our experiments, we use GPT-4.1 as the default verifier; however, we also examine the performance of GridAR (*N*=4) with different verifier choices. As shown in the centered plot of Figure 6, both GPT-4o-mini and the quantized Qwen2.5-VL-72B (8-bit) lead to consistent improvements - by 18.6% and 8.3%, respectively - on text-to-image generation. These results imply that our framework can further benefit from stronger vision-language models with enhanced image understanding capabilities. As this area continues to advance, we expect the upper bounds of our framework to rise with the emergence of more capable visual reasoning models.

**Effect of prompt reformulation timing** A natural research question is whether reformulating the prompt *before* image generation using a planner model could offer advantages over our strategy. While plausible, our approach instead performs prompt reformulation *midway*, using layouts inferred from partial images. To compare, we generate images directly from our reformulated prompt (Figure 6, right). While GridAR (*N*=4) with initial prompt reformulation achieves an average score of 0.5018 in text-to-image generation - showing a clear improvement over no reformulation (average score 0.4753) - our layout-aware reformulation yields a substantially higher score of 56.43. We attribute this to initial layout-guided prompts being susceptible to divergence from the intended layout, as early-stage next-token predictions may deviate from the structure.

## 5 CONCLUSION

We have introduced *GridAR*, a test-time scaling framework that rethinks how computation should be allocated in visual autoregressive (AR) models. Through progressive, grid-based generation and dynamic prompt reformulation, our approach selectively amplifies promising candidates while pruning suboptimal ones early - addressing limitations of conventional Best-of-*N* strategies. Without requiring training, *GridAR* draws out the full potential of AR models, achieving higher quality in both text-to-image generation and image editing. We believe this work marks a milestone for generative AR models, pushing the boundary of what test-time scaling can achieve in AR image generation.

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

## THE USE OF LARGE LANGUAGE MODELS (LLM)

We used Large Language Models (LLMs) only as a general-purpose assistive tool to improve the clarity of our writing and to generate alternative phrasings. The LLMs did not contribute to the research ideas, methodology, experiments, or analysis. All scientific content, results, and conclusions are entirely our own work.

## A  RELATED WORKS

**Visual autoregressive models**  Visual autoregressive (AR) models have recently advanced rapidly as a natural extension of the language-modeling paradigm to vision (Tian et al., 2024; Team, 2024; Sun et al., 2024; Chen et al., 2025). By adopting vector-quantized image representations, these methods discretize images into codebook indices, allowing the next-token prediction process to operate over image tokens. In practice, their generative quality is now competitive with strong diffusion baselines such as DALL·E 3 (Betker et al., 2023) and Stable Diffusion 3 (Esser et al., 2024).

Standard visual AR models decode in a raster-scan order (top-left to bottom-right), though several notable variants have been proposed. Masked autoregressive models (MAR (Li et al., 2024)) offer a unified formulation of standard AR models and masked generative models, while VAR (Tian et al., 2024) applies next-scale prediction instead of next-token prediction. Nevertheless, the standard raster-scan AR formulation remains a competitive and versatile baseline for both image understanding and generation, and has also been adapted for image editing (Mu et al., 2025).

**Test-time compute scaling**  While large language models (LLMs) have benefited significantly from test-time scaling techniques, such strategies remain underexplored in the context of visual autoregressive (AR) generation. In the language domain, increasing inference-time computation has proven to substantially improve model performance on challenging reasoning tasks such as mathematics (Wang et al., 2024a) and coding (Chen et al.), without any changes to model parameters. Two representative strategies include (i) chain-of-thought (CoT) prompting (Kojima et al., 2022), which extends generation length to encourage multi-step reasoning, and (ii) Best-of-$N$ sampling with outcome reward models (ORMs) (Snell et al., 2024), which samples multiple outputs and selects the best one based on task-specific reward signals. These methods demonstrate how allocating additional compute at inference can serve as a powerful tool to elicit more accurate or coherent outputs from pretrained models.

In contrast, such inference-time compute scaling remains largely unexplored in visual AR models, which face additional challenges such as spatial consistency, layout fidelity, and the lack of structured intermediate outputs. Unlike LLMs - where decoding follows a left-to-right sequence and allows for natural decomposition into steps - visual AR generation often proceeds in a fixed raster-scan order, offering limited flexibility for structured control or mid-generation interventions. Consequently, it remains unclear how to best allocate computation or incorporate structural feedback during the generation process for images. Our work addresses this gap by proposing a structured test-time scaling framework specifically tailored to visual AR generation, enabling dynamic pruning and refinement across intermediate steps without retraining.

## B  REJECTION ANALYSIS AND EXAMPLES

We investigated the rejection ratio at each stage of GridAR's grid-based progressive generation, where a grid is marked as *impossible* if rejected. In the first stage, one grid corresponded to one-quarter of the image ($R_1 = 4$), while in the second stage, each grid covered half of the image ($R_2 = 2$). Table 3 reports the proportion of rejected grids relative to the total number of generated grids at each stage, as well as the percentage of samples in which all grids were rejected. For Text-to-Image Generation, we used Janus-Pro, and for Image Editing, we used EditAR.

Across both tasks, rejections were rare in the first stage but occurred more frequently in the second stage. This pattern suggests that the verifier tended to pass grids when evidence for rejection was

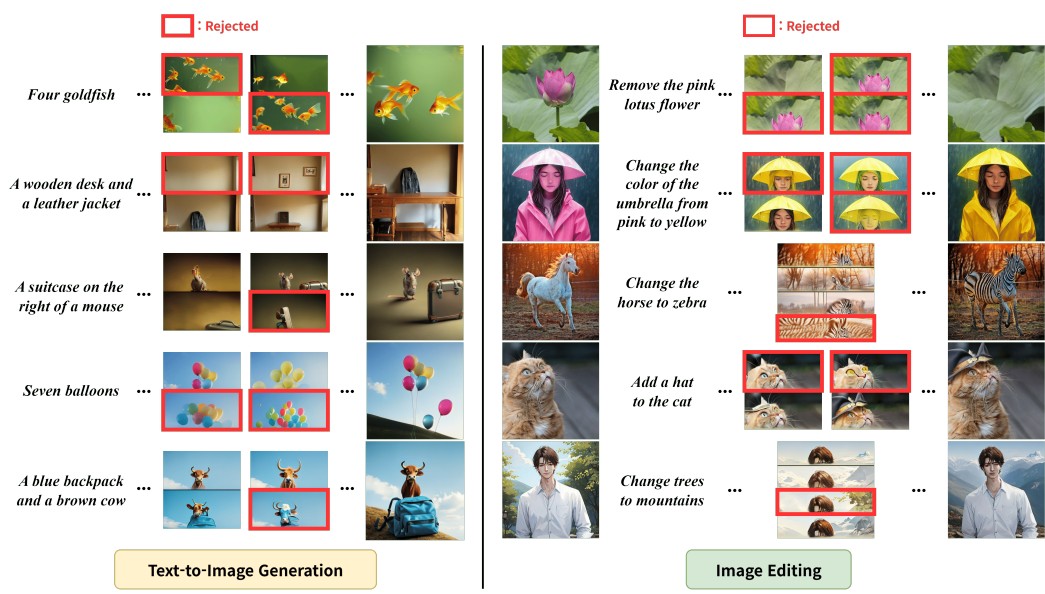

Figure 7: Examples of Rejected Samples.

Table 3: Rejection statistics by stage.

| Task | Model | Stage | Grid-level rejection rate (%) | Fully rejected sample rate (%) |
|---|---|---|---|---|
| **Text-to-Image Generation** | **Janus-Pro** | First-stage (R1=4) | 2.545 | 0.870 |
| | | Second-stage (R2=2) | 21.406 | 3.709 |
| **Text-to-Image Generation** | **LlamaGen** | First-stage (R1=4) | 10.753 | 1.271 |
| | | Second-stage (R2=2) | 34.305 | 7.167 |
| **Image Editing** | **EditAR** | First-stage (R1=4) | 28.545 | 20.429 |
| | | Second-stage (R2=2) | 43.252 | 21.286 |

definitely uncertain. Figure 7 illustrates qualitative examples for partially rejected candidates, which can also be seen in Figure 4 for Image Editing.

Cases where all grids within a sample were deemed impossible were infrequent (especially for text-to-image generation, which was extremely rare). In these situations, we continued the generation process rather than restarting from scratch, in order to maintain a fair comparison with Best-of-$N$ search, which would otherwise be disadvantaged by additional token generation costs. Nevertheless, in practical deployments of GridAR, we recommend different strategies depending on application requirements: for time-sensitive settings, selecting the top-$k$ grids may be more suitable, whereas in scenarios where performance is important, rolling back and regenerating from scratch may offer a more effective alternative.

## C DETAILED EXPERIMENTAL SETUP

**Baselines** We primarily compare against a simple Best-of-$N$ (BoN) baseline driven by an outcome reward model (ORM). To contextualize text-to-image performance, we report results alongside state-of-the-art diffusion systems (SDXL (Podell et al., 2024), PixArt-$\alpha$ (Chen et al., 2023), DALL·E 3 (Betker et al., 2023), Stable Diffusion 3 (Esser et al., 2024), and FLUX.1 (Labs, 2024)) and contemporary autoregressive generators (Lumin-mGPT (Liu et al., 2024), Emu3 (Wang et al., 2024b), Show-o (Xie et al.), and Infinity (Han et al., 2025)). For instruction-guided image editing, we benchmark diffusion-based editors—InstructPix2Pix (Brooks et al., 2023), InstructDiffusion (Geng

et al., 2024), MGIE (Fu et al., 2024)—and an autoregressive editor, EditAR (Mu et al., 2025), which generates tokens sequentially.

**Experimental Environment**  We conduct all experiments on 2x RTX 3090 GPUs. The batch size is set to 2. For a fair comparison with the Best-of-N (BoN) baseline, we ensure that the first row is generated identically across methods. Unless otherwise noted, the sampling temperature is fixed at 1.0.

**Evaluation Protocols**  Image editing is evaluated on PIE-Bench (Ju et al., 2024), which comprises 700 images across nine editing scenarios. Each sample includes a source image and a natural-language instruction, together with auxiliary annotations (source prompt, target prompt, and a ground-truth edit mask). Instruction following is measured via CLIP similarity (Hessel et al., 2021) between the edited image and the target prompt, computed over the full image and, separately, within the annotated edit region. Source preservation is assessed using a structure-aware distance derived from DINO-ViT (Caron et al., 2021) features ("structure distance") and standard fidelity/perceptual metrics—PSNR, SSIM (Wang et al., 2004), LPIPS (Zhang et al., 2018), and MSE.

# D  ADDITIONAL QUANTITATIVE RESULTS

## D.1  IMAGE EDITING

Table 4: Image Editing Results on PIE-Bench.

| Method | Structure Distance ($\downarrow$) | Background Preservation | | | | CLIP Similarity | |
|---|---|---|---|---|---|---|---|
| | | PSNR ($\uparrow$) | LPIPS ($\downarrow$) | MSE ($\downarrow$) | SSIM ($\uparrow$) | Whole ($\uparrow$) | Edited ($\uparrow$) |
| *Diffusion Models* | | | | | | | |
| InstructPix2Pix + BoN (N=4) | 54.903 | 21.054 | 146.089 | 216.340 | 77.361 | 24.917 | 22.271 |
| InstructDiffusion + BoN (N=4) | 76.023 | 21.066 | 131.994 | 308.671 | 77.375 | 24.760 | 22.126 |
| MGIE + BoN (N=4) | 70.417 | 21.502 | 135.859 | 304.363 | 78.190 | 24.792 | 22.405 |
| *Auto-Regressive Models* | | | | | | | |
| EditAR (N=1) | 41.580 | 20.721 | 125.214 | 205.538 | 74.079 | 24.813 | 22.092 |
| EditAR + BoN (N=2) | 40.890 | 21.065 | 118.810 | 166.652 | 74.648 | 25.103 | 22.247 |
| EditAR + BoN (N=4) | 40.948 | 21.368 | 113.829 | 140.533 | 75.043 | 25.386 | 22.436 |
| EditAR + BoN (N=8) | 42.837 | 21.464 | 111.252 | 135.404 | 74.979 | **25.628** | **22.626** |
| **EditAR + GridAR (N=4)** | **37.969** | **21.596** | **109.309** | **116.637** | **75.243** | 25.542 | 22.561 |

To adapt GridAR to the editing setting, candidate scoring is conditioned jointly on the source image and the text instruction. A grid hypothesis is retained only when both the instruction-following and source-preservation requirements are satisfied, in line with established image-editing evaluation practice (Ku et al., 2023). We further apply a single prompt reformulation when generation reaches the halfway point, after which selection proceeds under the same dual-criterion rule. Table 4 reports the quantitative results on PIE-Bench for all baselines, where all models are evaluated in the manner of Best-of-$N$ search, and the output reward model is aligned with the GridAR setting. Under the same $N$=4 configuration, GridAR emerges as the most effective test-time scaling method, achieving the best performance across all metrics. Moreover, for EditAR, GridAR delivers comparable CLIP similarity even against the Best-of-$N$ ($N = 8$) baseline, while providing consistently superior performance in terms of semantic preservation.

## D.2  ABLATION STUDY FOR TEXT-TO-IMAGE GENERATION

In Table 5, we present a stepwise evaluation to quantify the contribution of each component in *GridAR*, and to compare two prompt reformulation strategies: prompt replacement and our proposed 3-way CFG. Specifically, the Prompt Reformulation module refers to revising the prompt based on intermediate results during generation; Grid Generation denotes our progressive, grid-based image generation process; and 3-way CFG indicates our approach to guiding the model with orthogonal-ized reformulated prompts, rather than simply replacing the original prompt. As shown in the table, each component yields clear improvements, with the full combination achieving the best performance. The comparison between the second and third rows highlights the effectiveness of prompt

replacement, suggesting it as a cost-efficient option for practitioners when 3-way CFG is unavailable.

Table 5: Ablation study: Component-wise evaluation of *GridAR* ($N = 4$) on the T2I-CompBench++ benchmark with Janus-Pro.

| Prompt Reformul. | Grid Generation | 3-way CFG | Attribute Binding | | | Object Relationship | | | Numeracy | Complex |
|---|---|---|---|---|---|---|---|---|---|---|
| | | | Color | Shape | Texture | 2D Spatial | 3D Spatial | Non-Spatial | | |
| | | | 0.6781 | 0.4026 | 0.5305 | 0.2218 | 0.3024 | 0.7830 | 0.4958 | 0.3840 |
| ✓ | | | 0.7780 | 0.5620 | 0.6905 | 0.2470 | 0.3115 | 0.7690 | 0.5380 | 0.3752 |
| ✓ | ✓ | | 0.7837 | 0.5683 | 0.6969 | 0.2672 | 0.3418 | 0.7790 | 0.5487 | 0.3906 |
| ✓ | ✓ | ✓ | 0.8050 | 0.6014 | 0.7268 | 0.2833 | 0.3503 | 0.7887 | 0.5684 | 0.3905 |

# E    EXTENSIVE QUALITATIVE RESULTS

## E.1    TEXT-TO-IMAGE GENERATION

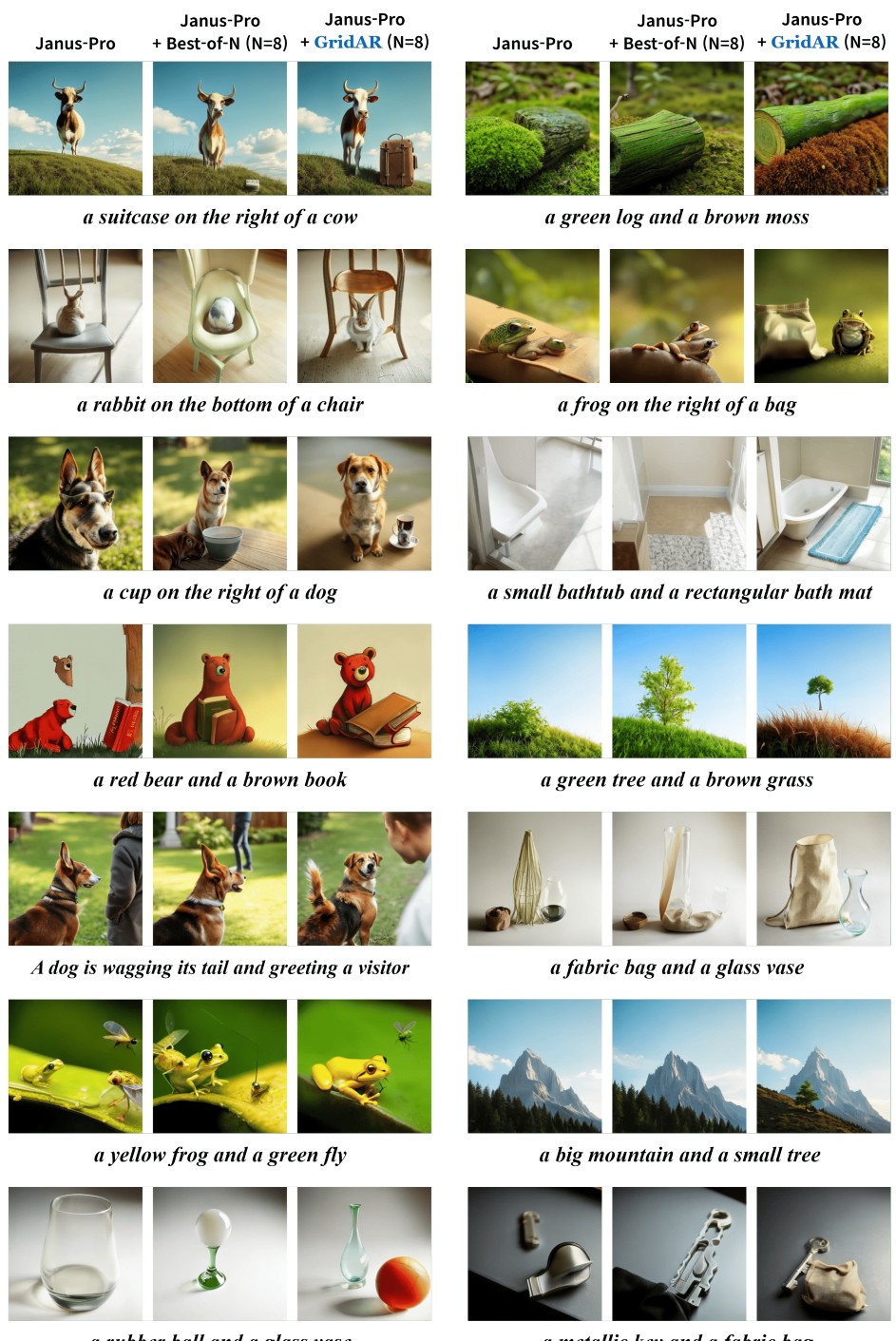

Figure 8: Additional Qualitative Results for Text-to-Image Generation

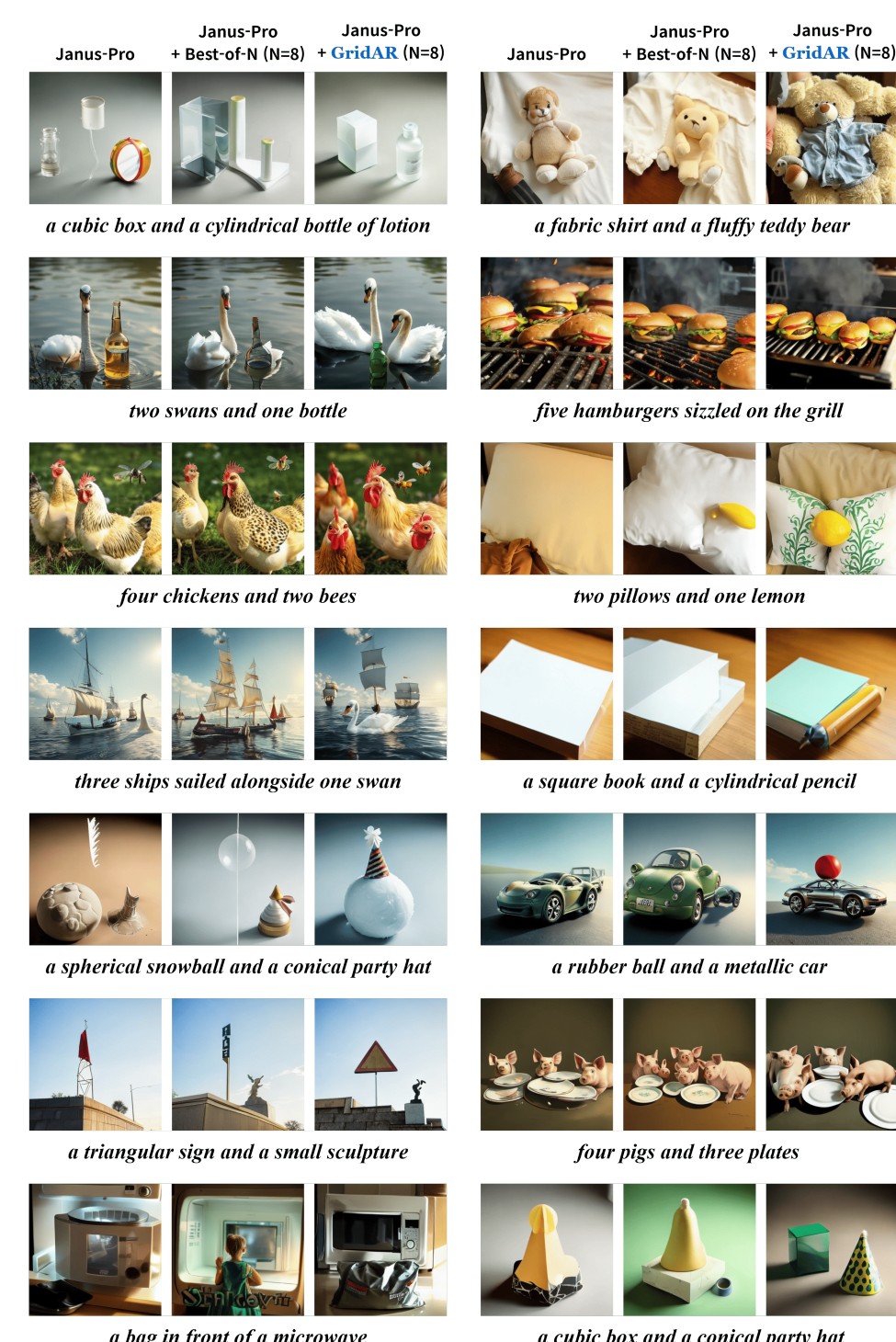

Figure 9: Additional Qualitative Results for Text-to-Image Generation.

## E.2 IMAGE EDITING

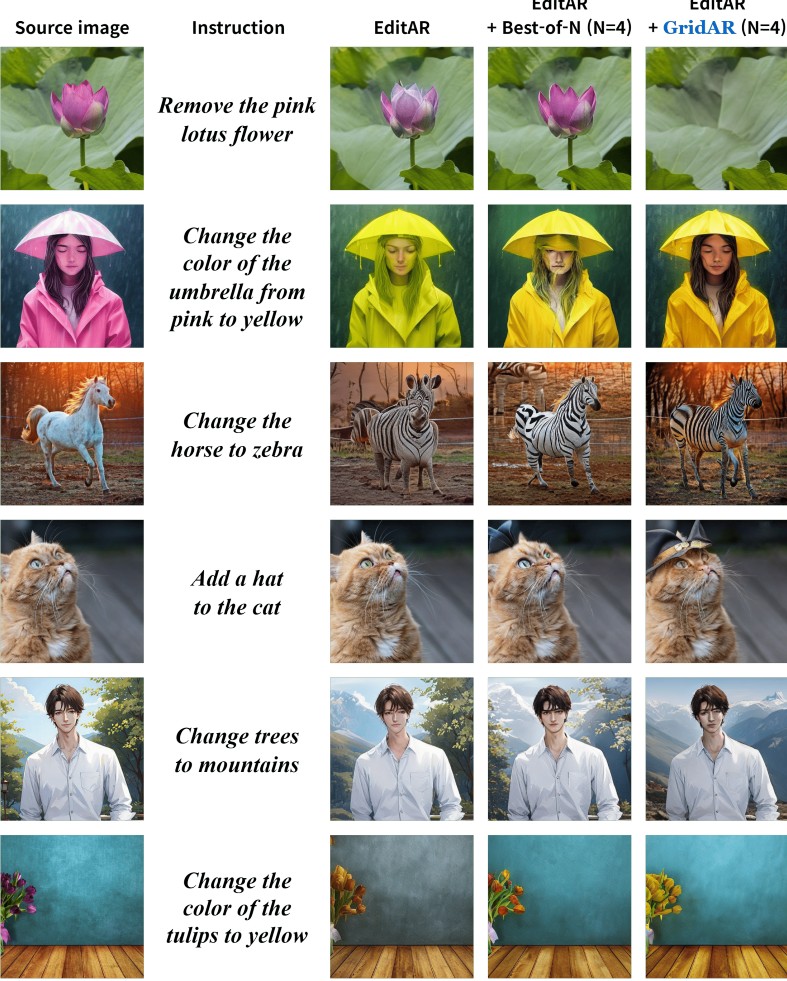

Figure 10: Additional Qualitative Results for Image Editing.

# F PROMPT TEMPLATES

## F.1 PROMPTS FOR TEXT-TO-IMAGE GENERATION

---

**Prompt for Selection of First Rows (Quarters) in Text-to-Image Generation**

You are given a single image consisting of 4 contiguous horizontal quarters (from top to bottom: quarter 1, quarter 2, quarter 3, quarter 4).
Each quarter shows the top quarter (upper 1/4 crop) of a different full image generated from the same text prompt. The lower three-quarters of each full image are not shown in this composite. Since only the top part is visible, some quarters may show only the background without any objects. In other cases, objects may appear only partially, with the rest continuing into the unseen lower part of the image.

The text prompt is: "{}".

For each of the 4 quarters, answer strictly with either "possible" or "impossible" (lower-case, no punctuation).
Output must contain exactly 4 words, separated by commas, in order: quarter 1, quarter 2, quarter 3, and quarter 4.
Example format: possible, impossible, possible, impossible

Focus on the required attributes (e.g., color, shape, counts, spatial relations) of objects in the prompt.

Say "impossible" for a quarter only if it is certain that the prompt cannot be satisfied:
- the visible part already makes it clear that the prompt cannot be satisfied (e.g., too many objects are already drawn, or an object has the wrong color or an incorrect shape), OR
- even if the lower three-quarters of that full image (not shown) were completed naturally, the final image would still not match the required attributes.

If there is any reasonable way the prompt could still be satisfied, say "possible".

---

**Prompt for Layout-Specified Prompt Reformulation in Text-to-Image Generation**

Rewrite the prompt "{}" considering the given partially generated images, so that it fully describes the final image layout with the correct total number of objects and accurately satisfies the original prompt, helping the model complete the remaining half.

RULES
- Keep the object type and total count exactly the same as in the original prompt.
- Do NOT add, remove, or change objects. Never alter the number.
- Do NOT introduce new attributes (colors, styles, or layout details) not present in the original prompt.
- Strictly preserve the original prompt at the beginning; only append a simple clause if it is directly useful (e.g., "<X> on the top and <Y> on the bottom").
- If the visible half already contains all required objects but only in incomplete form, leave the prompt unchanged; however, if the empty lower area could make the model add extras, append a minimal placement clause that locks the existing set (e.g., "centered in a single horizontal row").

OUTPUT
- Output exactly one sentence: either the refined prompt or the unchanged original.
- Begin with the original text prompt; when refining, append the clause after a comma.

EXAMPLES
Original: "A photo of eight bears"; Visible: three bears on the top →

---

Output: "A photo of eight bears, three on the top and five on the bottom."

Original: "A photo of one chicken"; Visible: only the upper half of the same chicken →
Output: "A photo of one chicken" (unchanged; continuation only)

...

## F.2   PROMPTS FOR IMAGE EDITING

**Prompt for Selection of First Rows (Quarters) in Image Editing**

You are given two images:
(A) the source image to be edited, and
(B) a composite image consisting of 4 contiguous horizontal quarters (from top to bottom: quarter 1, quarter 2, quarter 3, quarter 4).
Each quarter in (B) shows the top quarter (upper 1/4 crop) of a different full image produced by applying the same editing instruction to the source image. The lower three-quarters of each full image are not shown. Since only the top part is visible, some quarters may show only the background without any objects, while others may show objects partially, with the rest continuing into the unseen lower part of the image.

The editing instruction is: "{}".

For each of the 4 quarters, make two independent judgments:
1) instruction_following — does the visible part allow the edited image to plausibly satisfy the instruction?
2) source_preservation — do the visible regions unrelated to the instruction look reasonably consistent with the source image?

OUTPUT FORMAT (STRICT):
Return a single-line JSON object with exactly two keys:
{{"instruction_following": "<q1,q2,q3,q4>", "source_preservation": "<q1,q2,q3,q4>"}}
- For each key, the value is a lowercase string of exactly four words, each either "pass" or "fail", separated by a comma and a single space, in order: quarter 1, quarter 2, quarter 3, quarter 4.
- Example: {{"instruction_following": "pass, fail, pass, fail", "source_preservation": "pass, pass, fail, pass"}}
- No extra keys, punctuation, notes, or explanations.

Focus on whether the editing instruction applied to the source image could plausibly be satisfied given each quarter candidate, and separately whether areas not related to the instruction look reasonably consistent with the source image—keeping in mind that the limited quarter view alone is not evidence of failed preservation.

Guidance for instruction_following:
- Say "fail" only if the visible part clearly shows that the instruction cannot be satisfied (e.g., added/edited elements are missing or incorrect, or edits strongly contradict the instruction), OR
- even if the unseen lower part were completed naturally, the final image would still definitely not satisfy the instruction. If there is any reasonable way the instruction could still be satisfied, say "pass".
- Cropping that hides the edit is not a failure by itself.

Guidance for source_preservation:
- Say "fail" only if the visible part clearly breaks preservation of the source image in areas not required by the instruction (e.g., obvious identity swap, major background replacement unrelated to the instruction, severe artifacts/structural distortions that contradict the source).
- The limited quarter view alone is not evidence of failure. Do NOT mark "fail" merely because

the visible area looks unchanged; preservation often implies minimal change.

---

**Prompt for Layout-Specified Prompt Reformulation in Image Editing**

Rewrite the editing instruction "{}" considering the source image and the given partially edited images, so that it guides completion of the edit while preserving the source content and accurately satisfying the original instruction, helping the model complete the remaining half.

RULES
- Keep the core intent of the original editing instruction unchanged.
- If the instruction specifies object types, placements, or counts, keep them exactly the same; do NOT alter numbers or required entities.
- Do NOT introduce new attributes (colors, styles, layouts) that are not implied by the original instruction or the visible evidence.
- You may add minimal placement/scope phrasing only when directly supported by the visible evidence (e.g., "near the top edge", "lower area").
- Preserve unedited regions of the source image; it is allowed to explicitly say that non-edited areas must remain unchanged.
- UNCHANGED CASE: If the visible portion already satisfies the instruction (with only incomplete crops of the same edits), return the original instruction verbatim.
- OTHERWISE: Rewrite the instruction from scratch as a single imperative sentence that integrates the visible evidence; do NOT prepend, quote, or copy the original wording.

OUTPUT
- Output exactly one sentence.
- If unchanged, output the original instruction verbatim.
- Otherwise, output a single rewritten sentence (no quotes) that integrates visible cues and preserves counts/scope.

EXAMPLES
Original: "Change the animal from a cat to a dog"; Visible: only the upper half of a dog already replacing the cat →
Output: "Change the animal from a cat to a dog" (unchanged; continuation only)

Original: "Change the color of the horse from white to golden"; Visible: the horse's head and mane already appear golden while the body remains white →
Output: "Change the horse's body, legs, and tail to gold."

...

