# OpenReview forum: "GridAR: Glimpse-and-Grow Test-Time Scaling for Autoregressive Image Generation"
_ICLR.cc/2026/Conference — ICLR 2026 Conference Withdrawn Submission_

### Official Review · Reviewer_skzL · 2025-10-25

**Soundness:** 3
**Presentation:** 3
**Contribution:** 3
**Rating:** 8
**Confidence:** 4

**Summary:**

One of the most straightforward test-time scaling strategies for generative models is the Best-N method, where the model generates N (ideally diverse) outputs and then selects the best one using an internal or external verifier. The authors first highlight the limitations of this naive Best-N strategy in the context of autoregressive image generation: it expends full computational effort on erroneous generation trajectories, and the raster-scan decoding scheme lacks a global blueprint of the canvas, limiting the scaling benefits since only a few prompt-aligned candidates are produced.

To address these issues, they propose GridAR, a method that partitions the canvas into grids and progressively generates content. Additionally, the authors design specialized prompt formulations to guide GridAR generation, mitigating the lack of global structure typically observed in autoregressive models.

**Strengths:**

1 - Their strategy is conceptually simple yet powerful. They smartly adapt the Best-N approach to account for the limitations of autoregressive (AR) models, rather than applying it naively.

2 - They conduct a comprehensive set of experiments across two AR model families, Janus-Pro and LlamaGen, covering both image generation and editing tasks. Moreover, they emphasize benchmarks that demand substantial reasoning capabilities.

3 - I particularly appreciate the authors’ transparency in including diffusion model baselines in their benchmarks, providing a fair and holistic comparison.

**Weaknesses:**

1 - In Appendix F, their prompt templates include several dataset-specific details. While this is not a major concern, I’m curious about the robustness of their method when the amount of human prior knowledge in the prompt formulation is reduced. It would be valuable if the authors could provide a simple experiment using a less “engineered” prompt to demonstrate how performance changes under weaker prompt guidance.

2 - I found the computational analysis somewhat unclear. The authors mention using 2 RTX 3090 GPUs, yet they also state that GPT-4.1 was employed as the internal verifier. How is the computational footprint of GPT-4.1 accounted for in their analysis? It might be more transparent to use an open-source verifier, such as Qwen (already used in Section 4.4), to ensure a fair and reproducible evaluation.

3 - The authors devote a substantial portion of the paper to motivating and describing how classifier-free guidance (CFG) can be integrated into their test-time scaling strategy, referred to as “Three-way CFG.” However, it remains unclear whether the reported results actually incorporate this technique. Furthermore, the paper lacks an ablation study comparing Three-way CFG with the prompt replacement method, which would help clarify its contribution to overall performance.

**Questions:**

1 - Their grid-based strategy involves two stages: in the first stage, they use an R1-row grid to explore initial partial candidates, followed by an R2-row grid for continued generation from the selected anchor candidates. I’m curious why defining a separate R2 is necessary—couldn’t the same process be repeated using the original R1 rows while conditioning on the already generated tokens? It would be helpful if the authors could elaborate on the rationale or underlying intuition behind introducing R2, as understanding this design choice could clarify the overall philosophy of their approach.

---

### Official Review · Reviewer_2uLY · 2025-10-28

**Soundness:** 3
**Presentation:** 4
**Contribution:** 2
**Rating:** 4
**Confidence:** 4

**Summary:**

The paper proposes a test-time scaling strategy (GridAR) for raster-order autoregressive image generative models. Instead of Best-of-N, GridAR generates top-quarter candidates, uses a verifier to reject impossible ones, then expands to half-image candidates and repeats. Additionally,  layout-specified prompt reformulation conditions later decoding on the layout derived from the partials. GridAR improves over Best-of-N on text-to-image generation and image editing benchmarks.

**Strengths:**

1. The paper is clearly written and easy to follow.
2. Consistently gains on both T2I and editing benchmarks.
3. It is competitive to Best-of-N with larger N, while having lower processing time.
4. Verifying four partials in a single composite is an interesting idea to reduce the overhead.
5. The pilot study provides a convincing motivation for the layout-aware reformulation.

**Weaknesses:**

1. The claims are framed for visual AR models in general, but the method and experiments are grounded in raster-scan decoding (limited scope). It's not clear whether the approach extends to other common AR image generative models (e.g., Infinity, FlexTok).

2. The choice of ($R_1$, $R_2$) = (4,2) is adopted without ablation. If more compute is available, should we increase R (e.g., 8 parts, more verification steps) or simply increase N?

3. The evaluations focus on short compositional prompts (T2I-CompBench++, GenEval). No experiments are shown on long detailed prompts (e.g., DPG-Bench), where layout planning and the dependence on the early partials (first half) could become more challenging.

4. The method verifies four partial images at once by composing them into a grid, but the effect of this compositing on verifier accuracy isn't analyzed. If the verifier isn't perfect in this setting, showing some failure cases would be informative.

5. A broader discussion of limitations and failure modes of the method is missing.

Minor (not affecting the score): There is a color mismatch in Fig. 6 between the rightmost plot and its legend.

**Questions:**

1. Why is GPT-4.1 used for partial verification but Qwen used as the final ORM? Why not use GPT-4.1 for both? Since Best-of-N uses only Qwen, it would be informative to report the performance when using only GPT-4.1 as well (Fig. 6 suggests GPT-4.1 may perform better).

2. The grid method is presented first and then combined with prompt reformulation. It would help to include the performance of the grid-only method (without prompt reformulation) in Table 5 to isolate the contribution of each component.

3. What prompt template is used for the final ORM selection for both Best-of-N and GridAR? Only the mid-level prompts seem to appear in the appendix.

4. Have the authors tested prompt refinement on top of Best-of-N? For example, after generating the first quarter/half, use GPT-4.1 to derive a layout-guided reformulated prompt and continue generation, without using the rejection mechanism.

5. To confirm: in Layout-Specified Prompt Reformulation (L.1059), does the model output multiple layout sentences (one per partial image in the composite), or a single global sentence that reflects the layout across all partials?

6. Since the method does not explicitly rank candidates (only "impossible" paths are rejected), could the approach be used to find higher-quality candidates rather than just discarding incorrect ones? In other words, could this be used with a verifier with a continuous score?

---

### Official Review · Reviewer_f98B · 2025-10-30

**Soundness:** 3
**Presentation:** 4
**Contribution:** 3
**Rating:** 4
**Confidence:** 3

**Summary:**

They introduce GridAR, a test-time scaling framework for visual autoregressive (AR) models in text-to-image generation and editing. GridAR addresses the limitations of naive strategies like Best-of-N, which waste computation on early erroneous trajectories. It partitions the canvas into a grid and progressively generates multiple candidates per region, rejecting invalid candidates early while allowing viable ones to serve as anchors that guide subsequent decoding. To mitigate the blueprint deficiency inherent in raster-scan generation, GridAR incorporates layout-specified prompt reformulation, using partial views of the canvas to infer feasible layouts that guide the model in producing prompt-aligned images.

**Strengths:**

Novelty: GridAR introduces (1) a grid-partitioned progressive generation framework that adapts LLM-style test-time scaling to autoregressive image synthesis and (2) the proposed 3-way CFG disentangles original intent from layout-specific signals via orthogonalization, enabling precise steering without retraining.

Addresses core limitations of raster-scan decoding: Effectively mitigates the lack of a global canvas blueprint and the irreversibility of early errors by enabling early pruning of infeasible trajectories and anchor-guided continuation, directing computation toward viable paths and preventing resource waste on doomed generations.

Effect of Prompt Reformulation through experiment: Mid-generation injection of layout knowledge via prompt reformulation significantly boosts success rates over naive test-time scaling (Best-of-N), demonstrating the critical role of global planning in autoregressive image synthesis.

Scalable and modular design: The framework naturally scales to higher N (e.g., N=8 via two parallel starting canvases) and supports plug-and-play modularity

**Weaknesses:**

No Human Evaluation: Results rely on automated metrics; a user study on perceived quality or fidelity could strengthen claims, especially for subjective aspects like aesthetics in T2I or naturalness in editing.

Potential Overfitting to Benchmarks: Strong results on specific datasets, but generalization to real-world, diverse prompts (e.g., long-tail or ambiguous) isn't addressed.

Engineering: The fixed grid configuration (R1=4, R2=2) is an engineering choice without comparison to alternatives or adaptive strategies. (Lack of ablations later explained)

The rejection process can be noisy: Candidate rejection is binary and based on partial views, risking premature discarding of viable paths due to incomplete object visibility.

Rejection replacement strategy is simplistic: Rejected candidates are replaced by random duplication of accepted ones, which may reduce diversity and introduce bias.

No discussion of failure cases or error analysis: The paper shows success examples but lacks a systematic breakdown of when and why GridAR fails.

Lack of enough ablations: (1) No ablation on grid configuration, (2) No ablation on guidance scale tuning (3) No ablation on verifier prompt design (4) No ablation on rejection threshold (5) No ablation on image resolution or aspect ratio etc.

No Comparison to Other Test-Time Scaling Paradigms: GridAR is benchmarked only against Best-of-N, with no evaluation against other visual AR-specific paradigms such as token-wise Chain-of-Thought (CoT) or iterative masked modeling with verification, limiting claims of general superiority in test-time scaling strategies.

**Questions:**

1. How does varying the grid partition (e.g., R1=8, R2=1 or column-wise instead of row-wise) affect performance and cost? Are there experiments showing optimality of the chosen (4,2) setup?
2. Could the framework integrate with other scaling techniques, like chain-of-thought prompting for visual AR or hybrid AR-diffusion models?
3. How scalable is GridAR to larger N (e.g., 16+)? Does the rejection rate or verifier overhead become a bottleneck?
4. For image editing, how does GridAR handle instructions that require global changes (e.g., style transfer across the entire image) or non-localized edits, where partial views might not provide sufficient cues?
5. Is there a self-verification mode using the AR model’s own confidence or entropy?
6. Results appear as single numbers with no ±std or multiple runs, how stable are the gains?
7. Why is the gap in the non-spatial column in Table 1 between GridAR (N=8) and BoN (N=8) small?

---

### Official Review · Reviewer_2okJ · 2025-11-01

**Soundness:** 3
**Presentation:** 3
**Contribution:** 2
**Rating:** 4
**Confidence:** 5

**Summary:**

This paper presents GridAR, a novel test time scaling framework for visual autoregressive image generation. GridAR partitions the image canvas into a grid and progressively generates multiple partial candidates in parallel. Unlikely candidates are pruned early, while viable ones are kept to guide decoding. The key innovation is layout specified prompt reformulation (inspecting intermediate partial outputs and rewriting the prompt to be a reasonable layout fixing the lack of a “blue print” in AR models), the experiments show that this method significantly outperforms standard best of N sampling at a similar cost.

**Strengths:**

- The introduction of a grid structured progressive generation strategy (which is well motivated for AR image models), exploring candidates row wise and pruning early.
- The paper provides strong quantitative results. Across diverse prompt standardized benchmarks, also ablations showing each component adds a clear benefit.
- Beyond T2I, this model is also applied to image editing and yielding better instruction following and preservation on simple scaling baselines.

**Weaknesses:**

- The approach assumes the zero shot verifier rarely makes mistakes. If a correct partial candidate is wrongly classified as “impossible”, it is pruned. There is not enough analysis on this risk.
- The paper focuses on successful cases, lacking examples or stats of when the model fails (e.g. prompt reformulation leading to worse outcomes).
- The verification based pruning step is effectively a BoN style cherry picking performed earlier in the trajectories, but with added cost (partial decoders and VLM calls). The paper argues this focuses compute on promising branches yet it does not convincingly show that their method is more sample efficient that a carefully tuned BoN (or any other alternative resampling for that matter) at comparable latency.
- The method has 2 levers in effect, one is the pruning and the other is layout aware prompt reformulation. But there is partial attribution only; missing grid-only ablation and compute-matched seed-averaged analysis.
- Table 1 reports several vanilla diffusion backbones but omits diffusion methods that also do test time scaling/selection/optimization (e.g CARINOX-style reward-guided noise optimization + exploration see https://arxiv.org/abs/2509.17458). As a result, the diffusion rows serve mostly as reference points rather than true baselines, making the AR-vs-diffusion contrast misleading. If the goal is to position GridAR against the broader state of inference-time scaling, include compute-matched diffusion systems that explicitly spend extra test-time compute, not only single-pass models.
- There is no clear mention or direction of future works (only mentioning essentially bigger base models) or flaws of this method in the conclusion or anywhere else in the paper.
- The paper asserts a more favorable cost–performance trade-off based on a single Pareto slice. However, there’s no thorough, compute-matched study across (i) varying N, (ii) grid configurations, and (iii) verifier choices. Crucially, the overhead of verification (number/timing of VLM calls, GPU idle time while waiting on the verifier, partial-decoding cost) isn’t broken down. As a result, it’s unclear whether the pipeline is truly more sample efficient than a tuned Best-of-N (or alternative resampling) at the same wall-clock and same accelerator budget.

Formatting Issues:
Figure 6 Right legend color for “GridAR (N=4) + Initial Reformulation” is red but in figure it is purple
Figures 3, 4, 7, 8, 9, and 10 are not pdf

**Questions:**

- Can you quantify false-negative rates of the zero-shot verifier on partial tiles (and provide a sensitivity analysis over thresholds/temperatures) to show how often feasible candidates are wrongly pruned and how that impacts final scores?
- What prompts or conditions cause GridAR to underperform (e.g., harmful prompt reformulation or pruning away the eventual best branch), and what are the aggregate frequencies with illustrative examples?
- At matched FLOPs, how does GridAR compare to a carefully tuned Best-of-N (or progressive resampling) across N, including full Pareto curves (score vs time/FLOPs) and a breakdown of decoding vs verifier/prompt-rewrite overheads?
- Can you provide a compute-matched, seed-averaged ablation isolating Base AR, Grid/Pruning-only, Prompt-rewrite-only, and Both, to quantify each lever’s contribution on T2I-CompBench++/GenEval (and PIE-Bench if applicable)?
- Will you add compute-matched inference-time-scaled diffusion baselines (e.g., reward-guided noise optimization/exploration) or explicitly state that the diffusion rows are reference-only and limit the paper’s claims accordingly—and include a brief limitations/future-work section?
- Can you run a compute-matched ablation across multiple verification models (e.g., GPT-4.1, GPT-4o-mini, Qwen2.5-VL, and a lightweight heuristic), reporting absolute scores, wall-clock/FLOPs, rejection rates, and mean ± std over ≥3 seeds per prompt set, plus a calibration/threshold sweep to quantify each verifier’s effect size and variance on pruning decisions and final outcomes?

---

### Note · Authors · 2025-11-14

**Comment:**

We sincerely appreciate the reviewers’ thoughtful and constructive feedback. We will incorporate their suggestions into the next revision. Thank you again for your time and effort.

**Withdrawal Confirmation:**

I have read and agree with the venue's withdrawal policy on behalf of myself and my co-authors.